# Do LLMs Understand Your Translations?
# Evaluating Paragraph-level MT with Question Answering

**Patrick Fernandes**[*♣♠♡], **Sweta Agrawal**[*♣], **Emmanouil Zaranis**[♣♠],
**André F.T. Martins**[♣♠♢], **Graham Neubig** [♡]
[♡]Carnegie Mellon University
[♣]Instituto de Telecomunicações
[♠]Instituto Superior Técnico, Universidade de Lisboa
[♢]Unbabel
pfernand@cs.cmu.edu

## Abstract

Despite the steady progress in machine translation evaluation, existing automatic metrics struggle to capture how well meaning is preserved beyond sentence boundaries. We posit that reliance on a single *intrinsic* quality score, trained to mimic human judgments, might be insufficient for evaluating translations of long, complex passages, and a more "pragmatic" approach that assesses how accurately key information is conveyed by a translation in context is needed. We introduce TREQA (Translation Evaluation via Question-Answering), a framework that *extrinsically* evaluates translation quality by assessing how accurately candidate translations answer reading comprehension questions that target key information in the original source or reference texts. In challenging domains that require long-range understanding, such as literary texts, we show that TREQA is competitive with and, in some cases, outperforms state-of-the-art neural and LLM-based metrics in ranking alternative paragraph-level translations, despite never being explicitly optimized to correlate with human judgments. Furthermore, the generated questions and answers offer interpretability: empirical analysis shows that they effectively target translation errors identified by experts in evaluated datasets. Our code is available at https://github.com/deep-spin/treqa.

## 1 Introduction

Machine translation (MT) has progressed significantly in recent years, with top systems achieving translation quality comparable to human-generated references at the sentence level for many high-resource languages (Kocmi et al., 2024). A large part of this progress is driven by equally striking improvements in automatic evaluation metrics, where sentence-level neural metrics such as COMET (Rei et al., 2022a) and MetricX (Juraska et al., 2023) play a crucial role in assessing and improving MT models (Peter et al., 2023; Agrawal et al., 2024).

Despite these advances, progress in MT and automatic evaluation beyond the sentence level has been slow and limited. Challenges persist both in developing systems that can translate full documents coherently and consistently and in designing evaluation metrics that can appropriately measure the accuracy of translated cross-sentence phenomena. While various factors contribute to this slow progress, including the field's difficulty in moving away from sentence-by-sentence translation (Post & Junczys-Dowmunt, 2023), we posit that a fundamental issue lies in the reliance on *intrinsic* quality scores that, while useful for assessing individual sentences, fail to capture the complex, multi-faceted nature of document-level translation. Furthermore, existing metrics like BLEURT and COMET that predict these scores are typically trained on a large amount of human quality judgments at the sentence level—accumulated over the years of WMT evaluations (Koehn & Monz, 2006; Kocmi et al., 2024). However, such large-scale manual data collection becomes impractical at the document level, limiting the applicability of intrinsic-based measures for broader translation tasks.

---

* Equal contribution.

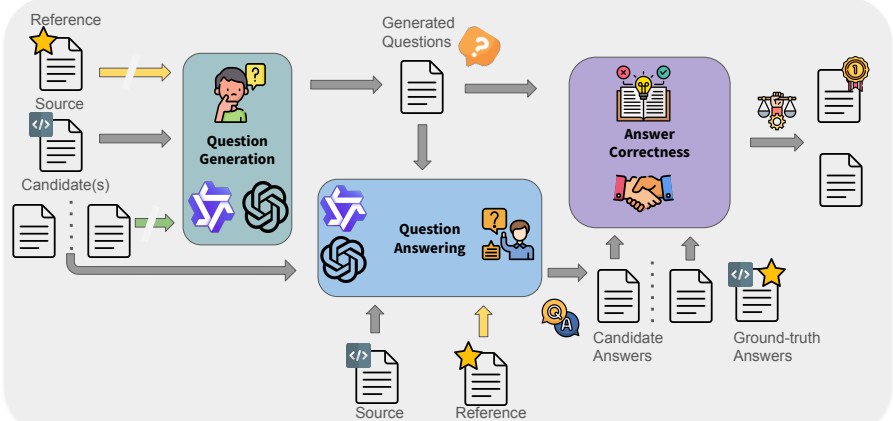

Figure 1: Approach overview: TREQA assesses translation quality through a reading comprehension question-answering framework. First, a set of questions $\mathcal{Q}$ is generated either in a candidate-aware (indicated by the green arrow) or a candidate-agnostic manner. Next, the QA system produces ground-truth and candidate answers. Finally, the Answer Correctness system evaluates and ranks candidate passages by comparing the candidate answers with the ground-truth answers obtained in the previous step. Access to reference passages is indicated by the yellow arrows.

Evaluating translations beyond individual sentences requires taking a more pragmatic approach that considers **how well a translation serves its intended purpose**—*i.e.*, how well it enables readers to use the information as effectively as readers of the source text. For example, readers of translated literary texts should get the same core information as if they were native readers of the original text. This type of *extrinsic* evaluation—where quality of the translation is tested through *reading comprehension questions* rather than merely predicting a single score—has been explored in prior works due to its potential to better measure the real-world utility of translations (Tomita et al., 1993; Fuji et al., 2001; Berkaab et al., 2011; Scarton & Specia, 2016; Forcada et al., 2018; Krubiński et al., 2021). However, most of these approaches were dropped in favor of the simpler judgment-based evaluation due to poor scalability since substantial human effort is required to generate questions and verify answers. Furthermore, questions were generated *uninformed* by the hypotheses being evaluated, making it necessary to exhaustively cover all key concepts to capture all errors.

We argue that, in the age of LLMs, the time is ripe to reconsider the usefulness of extrinsic evaluation for MT. In fact, LLMs can automate much of the human-intensive effort in question generation and answer verification steps, which has hampered previous approaches, as they have been shown to generate good questions to test comprehension in other natural language generation tasks (Eyal et al., 2019; Deutsch & Roth, 2021; Deutsch et al., 2021; Nan et al., 2021; Fabbri et al., 2022). In particular, we propose a question-answering-based automatic metric, TREQA (Translation Evaluation via Question Answering), which uses LLMs to generate questions about content in the source or reference texts targeting key concepts, phrases, and entities. To narrow down the space of possible questions and improve their effectiveness in localizing errors, we *optionally* condition the question generation on the set of hypotheses being evaluated. This allows the model to generate more targeted questions based on key differences between the hypotheses and the source (or the reference). We then assess translation quality by measuring how well answers extracted from the candidate translations align with those from the original source texts or references. Importantly, and unlike most state-of-the-art metrics, **TREQA is fully unsupervised: it does not require any training on human quality assessments dataset**.

We evaluate TREQA on literary texts – an especially challenging and underexplored domain for MT evaluation as these texts demand deep semantic understanding, multi-sentence reasoning, and subtle discourse-enforced meaning nuances that traditional metrics would fail to capture. We show that TREQA is competitive with or even outperforms state-of-the-art neural and LLM-based metrics in ranking alternative paragraph-level translations of literary texts translated into English from multiple languages, including Polish, Japanese, Russian, French, German, and Chinese, both in the reference-

based and reference-free (or *quality-estimation*; QE) setups. TREQA performs particularly better when question generation is conditioned on evaluated candidates – a setting that benefits from using more capable LLMs. While literature is open to interpretation, many questions generated by TREQA focus on elements such as character relationships, event sequences, and causal links, which have stable, answerable properties across translations. Furthermore, we demonstrate that the questions generated by TREQA precisely target expert-marked error spans, and generating these specific errors (rather than paraphrasing them) is critical for accurately capturing translation quality differences.

Nevertheless, despite their promise, the use of LLMs still pose challenges: we find that LLM-based evaluation can result in biased indicators of quality, such as position biases where generated answers favor candidates based on their order. We further find that variations in the generated question set can occur even with greedy decoding due to minor changes in low-level CUDA operations, causing evaluation instability. Interestingly, candidate-aware generation significantly reduces this variance by specifically targeting error spans and thus constraining the generation space, making evaluations more robust even when sampling. Finally, we discuss failure cases of TREQA, opening avenues for future research directions for improving controlled cross-lingual question generation and question answering (QA) and designing metrics for assessing answer correctness.

## 2 Translation Evaluation via Question Answering (TREQA)

**Formalization**    We aim to evaluate the quality of a candidate translation $\hat{y}$, given a source text $x$ and, optionally, a reference translation $y$. Drawing inspiration from prior research on using automatic QA-based metrics for evaluating MT (and other text generation tasks; Li et al., 2024), we hypothesize that, given relevant comprehension questions about the information conveyed in the source text $x$, a high-quality translation should allow a QA model to answer them perfectly.[1] Any deviations or errors in $\hat{y}$ (such as omissions, mistranslations, or distortions of meaning) can result in lower QA scores, either because the necessary information is missing or because the mistranslation introduces a mismatch in semantic content between $x$ and $\hat{y}$.

To quantify the quality of $\hat{y}$, we compare the answers derived from $\hat{y}$ (via a QA model) with the reference answers directly obtained from $x$ or a high-quality reference translation, $y$. Formally, given a *distribution* of possible questions $\pi_r$ that can probe the semantic content of a *ground-truth* text $r$ (source text $x$ and/or reference translation $y$), $\mathcal{M}(\hat{y}, r)$ is the quality score of $\hat{y}$ such that:

$$\mathcal{M}(\hat{y}, r) = \mathbb{E}_{q \sim \pi_r}\left[\rho\left(\text{QA}(\hat{y}, q), \text{QA}(r, q)\right)\right] \approx \frac{1}{|\mathcal{Q}|} \sum_{q \in \mathcal{Q}} \rho\left(\text{QA}(\hat{y}, q), \text{QA}(r, q)\right),$$

where, $\mathcal{Q}$ is the set of questions sampled from $\pi_r$ using an automatic question-answer generation (QAG) model; $\text{QA}(\hat{y}, q)$ is then the QA model's answer to the question $q$ using the candidate translation $\hat{y}$; and $\rho(\hat{a}, a)$ is a comparison function that measures the similarity between the QA model's answer, $\hat{a}$ (derived from $\hat{y}$) and the reference answer, $a$ (derived from $y$ or $x$). Together, these comparisons serve as a proxy for measuring how well the translation preserves the source text's meaning.

However, we need to make several design decisions to implement this metric effectively: the choice, granularity, and coverage of questions generated by the QAG model matter substantially, as irrelevant or poor-quality questions may fail to probe critical aspects of the semantic fidelity between $\hat{y}$ and $x$ or $y$. Relatedly, the use of capable QA models that can understand the content of the text and the questions, as well as the choice of comparison functions that can accurately measure the semantic similarity between the answers are crucial. We next discuss these components in detail.

### 2.1 Question-Answer Generation

The quality of the evaluation depends heavily on the questions used to probe semantic accuracy. We hypothesize that good questions should ideally be *discriminative*—able to distinguish between good and bad translations—and *should focus on concepts salient to the context*, where meaningful divergences between translations are likely to occur, while disregarding superficial (easy) details.

---

[1] This is assuming access to a "perfect" QA which simplifies the theoretical framework, however, in practice, QA models are imperfect and pose many limitations (Kamoi et al., 2023).

LLMs are particularly well-suited tools for generating such targeted questions. Their broad knowledge and reasoning capabilities allow them to identify important information and formulate probing questions that focus on key semantic elements. And, prior work has demonstrated LLMs' effectiveness in generating discriminative questions for monolingual text generation tasks like simplification (Trienes et al., 2024) and summarization (Kim et al., 2024b).

When *prompting* LLMs to generate question-answer pairs, which are then parsed into a set $\mathcal{Q}$, the choice of text to condition generation on plays a crucial role in shaping the types of questions produced. Importantly, we always generate both questions and answers, as opposed to just questions, to ensure that the questions are anchored in the provided text(s) and target verifiable information (Ushio et al., 2023). We consider two different settings:

**I. Candidate-agnostic QAG**: In this setup, the LLM generates question-answers based on the source text $x$ (making the system a *quality estimation* metric), the reference translation $y$, or both. The questions are generated in *"isolation"* without any information on the candidates being evaluated. This approach allows questions that target salient and contextually important concepts without being influenced by any particular translation and its potential errors, but might suffer from high variance (as the set of questions needed to evaluate all possible translation errors can be intractably large).

**II. Candidate-aware QAG**: We assume prior knowledge of the candidate translations to be evaluated. Using this information, we generate questions targeting potential errors grounded in the candidate translations and reference texts, $r$ (source and/or reference). To accomplish this, we prompt the LLM with the complete set of candidate translations.

Figure 2a shows the prompts used for each approach when both reference and source texts are provided as inputs (QE prompts are shown in Appendix 7). Although research on multilingual or cross-lingual capability of LLMs for question generation (when the source or reference is not in English) has been limited, some studies have suggested that fine-tuned QAG systems can generalize to unseen languages (Kumar et al., 2019). In this work, however, we constrain the generation of question-answer pairs to English. This choice is motivated by the fact that the capabilities of LLMs in English, nevertheless, are far superior to those in other languages, as evidenced by prior work (Etxaniz et al., 2023). Nevertheless, our setup still probes the cross-lingual understanding abilities of LLMs, given that the source or reference texts may be in other languages.

## 2.2 Question Answering

Given questions $\mathcal{Q}$, we face the challenge of selecting a QA model capable of answering them. Similar to question generation, LLMs are also well-suited for this task, with prior work demonstrating their effectiveness in QA-based evaluation (Fabbri et al., 2022; Xiao et al., 2023; Agrawal & Carpuat, 2024; Säuberli & Clematide, 2024; Trienes et al., 2024). Furthermore, while QG is more open-ended and requires generating diverse and meaningful questions, QA is a more constrained task that involves extracting relevant information from the text. Hence, while larger, more capable models may be necessary for high-quality QG, smaller, more efficient models could be sufficient for QA, provided they have strong reading comprehension abilities.[2] We focus on **extractive** QA, as this ensures that only information explicitly present in the text is considered rather than allowing a generative QA model to hallucinate plausible but incorrect answers (Rajpurkar et al., 2016; Mallick et al., 2023).

Note that the questions are always generated in English; however, the context could be English or non-English. To support both settings, we design two prompts: a) **monolingual**, ask the LLM to extract the answer from the context *as-is* and b) **crosslingual**, generate an answer given the cross-lingual context and English questions. Figure 2b shows our prompt(s) for answering questions using the candidate translation $\hat{y}$.

**Answer (Re)-Generation** While we prompt the LLM to generate both questions and answers, these answers are not used directly. Instead, they help scaffold high-quality, answerable questions (Ushio et al., 2023). These initial answers may differ stylistically or semantically from those produced by the QA model, even with access to the same text. Moreover, the QAG system can introduce bias by extracting answers from the candidate texts rather than the ground-truth reference (as confirmed by our analysis in Section 5). To address this, we *re-generate* answers using the QA model by providing

---

[2]We present an ablation analysis for using smaller models for QA in Appendix D.3.

| Prompt for Question Generation (with optional candidates) | QA Prompts for Monolingual and Crosslingual Contexts |
|---|---|
| **System:** You are an expert teacher tasked with designing reading comprehension questions.

**User:** Generate question-answer pairs to verify translation accuracy. Each answer should be a key phrase, concept, or entity from the original passage (source or reference) that is important in context and could help detect potential errors or mistranslations in the alternatives.
The questions and answers must be in English while preserving meaning. Questions should be diverse and cover different aspects of the passage.
Q: <question1>
A: <answer1>
Q: <question2>
A: <answer2>
**Source Passage:** {src_passage}
**Reference Passage:** {ref_passage}
Alternative Passages: {alternatives}
**Question-Answer Pairs:** | **System:** You are a helpful AI assistant skilled in extractive question answering.
*Monolingual setting*
**User:** Given the following passage and question, extract the exact answer from the passage. The answer should be a short span found verbatim in the passage.

*Crosslingual setting*
**User:** Given the following passage and question, return the answer in English using only the information from the passage. The answer should be concise and faithful to the original content.

###
**Passage:** {passage}
###
**Question:** {question}
###
**Answer:** |

Figure 2: Prompts used in the *reference-based* QAG setting (left) and in QA (right). Text in gray denotes optional components for candidate-aware QAG and is omitted in candidate-agnostic setups. QE prompt for QAG is shown in Figure 7. In the *monolingual* setting, the context (source, reference, or candidates) is in English, while in the *crosslingual* setting, the context is in some other language.

the question and reference (or source) text: $a = \mathrm{QA}(r, q)$. This aligns answer generation with the QA model's behavior and reduces bias, leading to more consistent and robust evaluation.

**Answer Correctness** We evaluate answer correctness using LERC (Fabbri et al., 2022), a learned metric that scores semantic similarity between a generated and reference answer, conditioned on the context and question, on a 1–5 scale. Trained on crowdsourced judgments of QA outputs, LERC has been widely adopted in QA-based summarization evaluation (Huang & Zhang, 2021; Deutsch & Roth, 2022) and outperforms surface-level metrics like exact match and word-level F1, which often fail to capture semantic equivalence.[3]

## 3 Experimental Setup

**Data:** We evaluate our metric on the following datasets, focusing on the xx-en translation direction:[4] a) **LITERARY Dataset**, released by Karpinska & Iyyer (2023) that includes MQM (Multidimensional Quality Metrics) error span annotation and expert preferences for automatically generated translations of paragraphs of literary texts. We use the XX-EN subset covering Polish, Japanese, French, German, Russian, and Chinese as source languages, retaining 180 instances out of 540. and b) **PAR3HUMAN Dataset** which includes expert preference between a Google Translate output versus a reference translation written by a human of public-domain foreign language novels for three language pairs (French-English, Russian-English, and German-English). We measure ***contrastive accuracy*** – how well the metric predicts expert preferences between translation pairs given either the source (reference-free) or both the source and the reference (reference-based).

**Baselines:** We compare TREQA against several baseline metrics: (1) lexical metrics such as CHRF (Popović, 2015); (2) sentence-level neural metrics like COMET (Rei et al., 2022a), COMETQE (Rei et al., 2022b), reference-based and reference-free versions of XCOMET (Guerreiro et al., 2024) and MTEQA (Krubiński et al., 2021) a reference-based QA metric that uses keyphrases as extracted answers; and (3) LLM-based metrics grounded in GEMBAMQM (Kocmi & Federmann, 2023a) using the same LLM backbone as TREQA. Further details on these metrics are provided in Appendix B.

---

[3]In our preliminary experiments with prompting LLMs for evaluating answer correctness, we found that they often fail to generate diverse correctness scores, with many answers scoring ties at 4.

[4]This choice is motivated by LLMs' superior performance in English compared to other languages (Etxaniz et al., 2024), the availability of a robust answer comparator (LERC) in English, and existing paragraph-level translation judgments for this direction.

| Metric | Model | *Cands?* | LITERARY (XX→EN) | | PAR3HUMAN | | |
| | | | *Ref* | *QE* | DE→EN | FR→EN | RU→EN |
|---|---|---|---|---|---|---|---|
| CHRF | — | — | 52.2 [2] | — | — | — | — |
| COMET | — | — | 62.2 [1] | — | — | — | — |
| COMETQE | — | — | — | 46.1 [4] | 37.3 [3] | 33.3 [2] | 51.3 [1] |
| XCOMET | — | — | 61.7 [2] | 55.6 [3] | 53.3 [2] | 41.3 [2] | 54.7 [1] |
| MTEQA | — | — | 43.3 [3] | — | — | — | — |
| GEMBAMQM | GPT-4O | — | — | 40.6 [4] | 44.0 [3] | 33.3 [2] | 52.0 [1] |
| | SONNET-3.5 | — | — | 51.1 [4] | **62.0** [1] | **54.0** [1] | **61.3** [1] |
| | QWEN2.5-72B-IT | — | — | 55.0 [3] | 55.3 [2] | 44.0 [2] | 55.3 [1] |
| TREQA | GPT-4O | ✗ | 59.4 [2] | 43.9 [4] | 44.0 [3] | 40.0 [2] | 54.0 [1] |
| | | ✓ | **65.0** [1] | 56.7 [2] | 46.7 [3] | 43.3 [2] | 56.0 [1] |
| | SONNET-3.5 | ✗ | 63.3 [1] | 56.1 [3] | 44.7 [3] | 42.7 [2] | 52.7 [1] |
| | | ✓ | 58.3 [2] | 55.0 [3] | **52.0** [2] | **47.3** [2] | 58.0 [1] |
| | QWEN2.5-72B-IT | ✗ | 64.4 [1] | **57.8** [1] | 44.7 [3] | 40.0 [2] | 57.3 [1] |
| | | ✓ | 64.4 [1] | 46.7 [4] | 45.4 [3] | 46.7 [2] | **58.7** [1] |

Table 1: Meta-Evaluation with Contrastive Accuracy on LITERARY and PAR3HUMAN, with both reference-based and reference-free (QE) metrics on the former and QE metrics on the latter. Methods highlighted with blue background use LLMs.

**LLM Backbones:** For TREQA (as well as GEMBAMQM) we experiment with a diverse set of state-of-the-art language models, including both the leading closed models and more reproducible open-weight options: for closed models, we explore both GPT-4O (Hurst et al., 2024) and SONNET-3.5 (Anthropic, 2024)[5], two of the leading closed-source models across many NLP tasks and that support the languages targeted. For open-source models, we experiment with the QWEN2.5-72B-IT and its smaller siblings QWEN2.5-32B-IT and QWEN2.5-7B-IT, and this family of models are generally competitive with closed-source models in multilingual settings (Yang et al., 2024).

**Decoding** As mentioned, for all these models, we use one of the prompts from Figure 2.[6] To ensure that TREQA is reproducible and produces consistent results, we use greedy decoding when comparing to other metrics, setting the temperature to 0.[7] However, due to the open-ended nature of the question generation and since some tasks obtain better results sampling from LLMs (Song et al., 2024) , we also explore the impact of sampling on the performance of TREQA with the QWEN2.5 models, by decoding with the recommend sampling parameters.[8]

**Significance Testing** We report clusters for metrics based on statistically significant performance gaps. Specifically, we conduct pairwise Wilcoxon signed-rank tests between all metric pairs within each group, where the dataset and the type of evaluation (Ref or QE) define groups. To account for multiple comparisons, we apply the Benjamini-Hochberg FDR correction to the resulting p-values. Based on the corrected p-values (thresholded at 0.05), we construct a significance matrix indicating which metric differences are statistically significant. This matrix serves two purposes: identifying metrics that are significantly better than others for ranking, and acting as a distance matrix for agglomerative clustering to group metrics with similar significance patterns, resulting in final ranking.

## 4   Results

Table 1 shows contrastive accuracy for TREQA and baseline metrics on all datasets.

**TREQA outperforms or is competitive with baselines in both reference-based and reference-free setups.** On the LITERARY dataset, TREQA achieves higher contrastive accuracy than lexical

---

[5]`gpt-4o-2024-08-06`; `claude-3-5-sonnet-20241022`

[6]While TREQA by default requires question conditioning on all candidates, we present an ablation on using single candidate to probe translation errors in Appendix D.2.

[7]We found that, even with greedy decoding, LLMs could produce different generations depending the specific environment setup. We use `vllm==0.6.2` to perform decoding.

[8]`temperature=0.7,top_p=0.8,top_k=20`

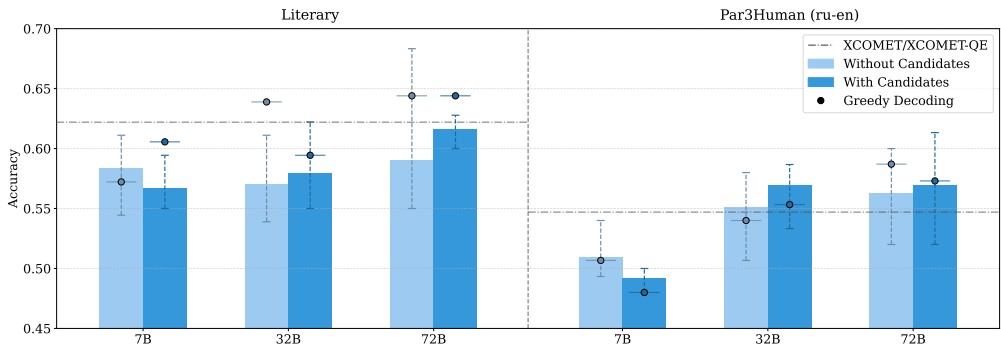

Figure 3: Mean, minimum, and maximum *accuracy* when sampling QA pairs 5 times with default sampling settings, as well as with greedy decoding and baseline performance, on LITERARY (reference-based) and PAR3HUMAN (RU→EN, reference-free) datasets.

and neural metrics, particularly when using strong closed-source LLMs like GPT-4O. Traditional lexical reference-based metrics, such as CHRF, perform at near-random levels. COMET and XCOMET achieve slightly stronger results but struggle to exceed 60%, while their QE counterparts perform even worse. Directly prompting LLMs to evaluate translations using GEMBAMQM also does not yield better results than state-of-the-art QE models like XCOMET, even with the best closed-source models. Whereas, TREQA consistently benefits from strong LLMs and outperforms all others on LITERARY when candidates are included, especially with SONNET-3.5 and QWEN2.5-72B-IT. On the PAR3HUMAN dataset, the picture is more nuanced: in the DE→EN direction, TREQA lags behind GEMBAMQM-based scores, and only marginally outperforms COMETQE. However, in the other two directions, TREQA proves a competitive evaluator, only being surpassed by GEMBAMQM with SONNET-3.5 (which is the best metric overall). Interestingly, in the FR→EN direction, except GEMBAMQM with SONNET-3.5, none of the other metrics surpasses random performance, hinting at the difficulty of current metrics in complex, long-form domains.

**There is no consensus on the best model for question generation, but candidate-aware generation generally outperforms candidate-agnostic.** Analysis across both datasets reveals that the optimal LLM for question generation varies: GPT-4O performs best on LITERARY, while SONNET-3.5 leads on PAR3HUMAN. In general, TREQA benefits from a candidate-aware setup, where the model sees the candidate translation when generating questions. However, there are exceptions—most notably, SONNET-3.5 achieves its highest performance on LITERARY in the candidate-agnostic setting. This discrepancy may stem from the inherent stochasticity of the question-generation process, an aspect we explore in more detail next.

**Sampling from larger models improves TREQA's accuracy over smaller models, and candidate conditioning improves mean and reduces variance.** As discussed earlier, the open-ended nature of question generation creates a vast space of possible QA pairs that can uncover different translation errors. Relying on a single deterministic generation (e.g., greedy decoding) may, therefore, underrepresent TREQA's full potential, as different questions expose different aspects of translation quality. To study this, Figure 3 shows the mean, minimum, and maximum contrastive accuracy of TREQA across five sampled runs—both with and without candidate conditioning, using various sizes of the Qwen family. Greedy decoding results and baseline performance from XCOMET are also included for comparison. As expected, accuracy improves with model size, with QWEN2.5-72B-IT consistently outperforming its smaller variants in both greedy and sampling settings. Comparing sampling to greedy decoding, the impact of candidate conditioning is mixed when looking at individual scores – no consistent pattern emerges across model sizes or datasets. However, when analyzing the distribution of outcomes across sampling runs, clearer trends appear: including candidates degrades performance for the smaller model but improves mean performance for the two larger models. This suggests that generating questions highlighting differences in candidate translations requires more capable models. Additionally, conditioning on candidates can make TREQA more stable, as shown by the low variance in accuracy of the QWEN2.5-72B-IT model on the LITERARY dataset. Finally, the ceiling of TREQA' performance is notably high, with one run achieving nearly 70% on the LITERARY dataset.

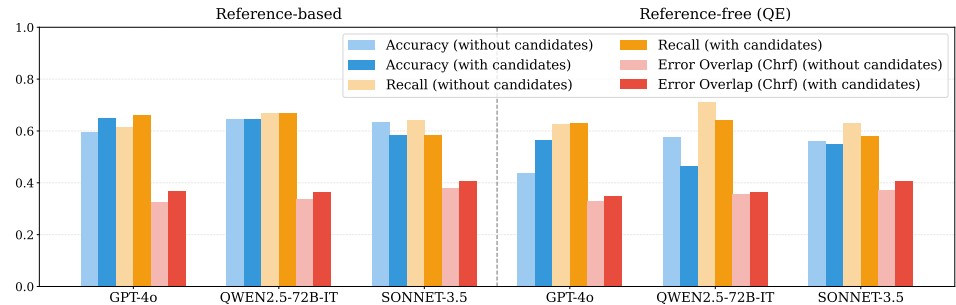

Figure 4: Accuracy, exact error coverage via recall and partial coverage (using CHRF) for the TREQA models presented in Table 1: recall reliably reflects accuracy trends.

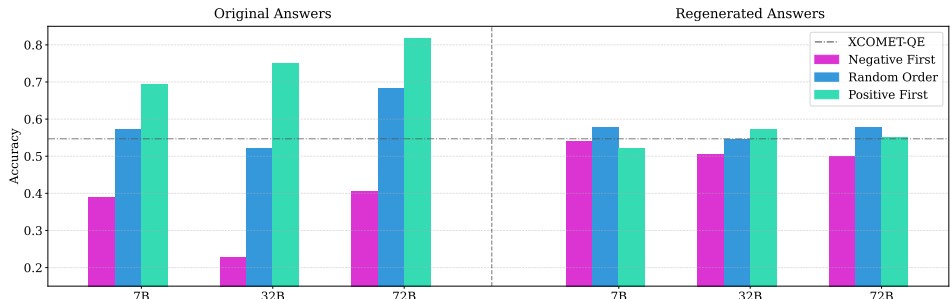

Figure 5: Impact on *accuracy* of presenting either the best candidate first, last, or in random order (Figure 2), using the originally generated answers (left) and regenerated (right).

## 5    Ablation Analysis

In this section, we examine how key design choices in TREQA affect its ability to generate meaningful and error-targeting questions, as well as the overall accuracy of candidate ranking.

**Do the generated questions target translation errors?** We hypothesize that candidate-aware generation enables the LLM to produce questions that specifically target translation errors. To test this, we evaluate the overlap between model-generated answers and expert-annotated error spans in the LITERARY dataset. We use two complementary metrics: (I) exact coverage via recall—whether the generated answer fully contains an annotated error span—and (II) partial coverage using CHRF (Popović, 2015), which quantifies the similarity between generated answers and error spans (see Appendix C for additional details). Exact recall offers a coarse-grained signal and may favor longer responses, while CHRF provides a finer-grained similarity measure. Figure 4 shows these metrics for the TREQA models from Table 1, comparing reference-based and reference-free setups with candidate-aware and candidate-agnostic question generation. Recall with error spans reliably reflects accuracy trends across models and settings. While candidate-aware generation consistently improves CHRF, exact recall better tracks overall performance. This is especially clear with QWEN2.5-72B-IT in the QE setup: CHRF improves with candidates but recall drops, suggesting the answers include related fragments or paraphrases without fully covering the error span. This observation highlights that identifying the entire error span is more critical than merely achieving surface-level similarity through partial matches or paraphrased content.

**How important it is to *regenerate* the answers with the QA system?** To assess whether answer generation is truly necessary, we compare accuracy using QAG-generated gold answers versus QA-generated ones. Figure 5 shows TREQA results when the best or worst candidate is presented either first, last, or in random order, and when using either the original or regenerated answers. We find that performance with QAG-generated answers increases sharply when the best candidate is presented first, and drops below random when the worst is first. This reveals a strong positional bias: when generating questions conditioned on candidates, LLMs often ignore instructions and instead

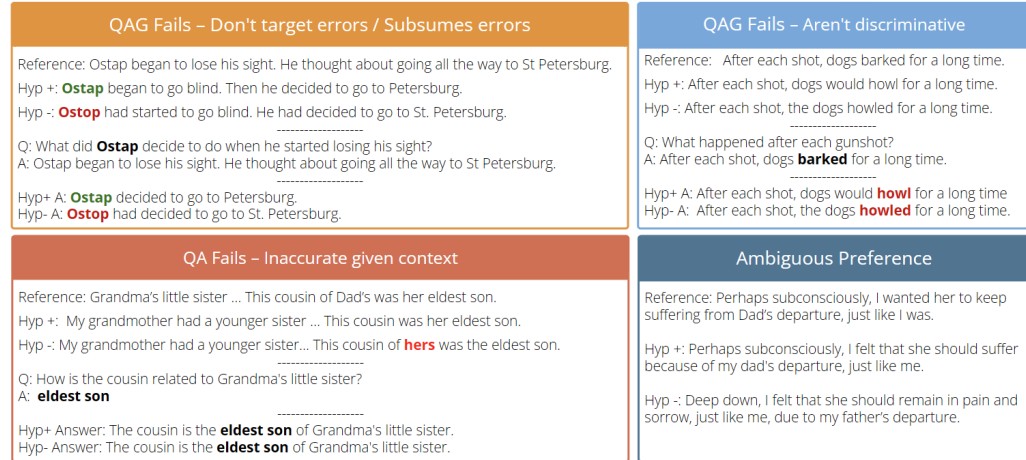

Figure 6: We highlight several failure cases of TREQA: questions include errors (top-left) or target same error for both candidates (top-right), QA generates answers not supported by context (bottom-left) and scoring fails when expert preferences are ambiguous (bottom-right).

copy answers directly from the first candidate—especially in the QE setting (see also Wei et al., 2024). This behavior undermines instruction-following and suggests shallow reasoning. Regenerating answers mitigates the issue by forcing the model to ground its output in the reference rather than copying from candidates, thus better reflecting true comprehension.

**Where does TREQA fail?** To better understand where TREQA falls short, we also manually analyze failure cases using the QWEN2.5-72B-IT model. We find that errors fall into three main categories: (1) generated questions fail to target errors or inherit candidate errors in the questions (Kamoi et al., 2023) or are not discriminative between candidates; (2) incorrect answers (paraphrased or inferred by the QA model) given the context (for ground truth or candidate) and (3) cases where the expert preference is ambiguous or underspecified, making reliable evaluation difficult. We provide examples of some of these categories in Figure 6. While TREQA is competitive with strong baselines and offers interpretable outputs, our analysis suggests several promising directions for future improvement. In particular, more controlled question generation targeting errors in a pragmatic way, improved QA systems, and better alignment in answer matching could further improve the reliability and precision of our metric, especially in challenging domains.[9]

# 6  Related Work

**Automatic Evaluation** Prior automatic evaluation methods apply sentence-level metrics directly to longer texts or explicitly adapt them for paragraph-level evaluation (Deutsch et al., 2023; Raunak et al., 2023; Vernikos et al., 2022), often assuming one-to-one sentence alignment. Such assumptions fail in realistic scenarios involving structural variations (sentence segmentation, reordering, restructuring). Discourse-level metrics (Joty et al., 2017; Jiang et al., 2022) address these by evaluating coherence through discourse categories. In contrast, TREQA uses LLM-derived answers to capture deeper semantic and pragmatic nuances beyond span-level alignments.

**QA-based Evaluation** QA-based evaluation methods have gained popularity across NLP tasks such as summarization (Durmus et al., 2020; Wang et al., 2020; Scialom et al., 2021; Fabbri et al., 2022; Liu et al., 2024), simplification (Angrosh et al., 2014; Agrawal & Carpuat, 2024), knowledge-based dialogues Honovich et al. (2021), long-form generation (Tan et al., 2024), and machine translation (MT) (Berkaab et al., 2011; Forcada et al., 2018; Krubiński et al., 2021; Han et al., 2022). Specifically, Krubiński et al. (2021) propose MTEQA, focusing on keyphrase-driven questions correlating strongly with human judgments at the system-level. Our method differs by employing

---

[9] We show the distribution of question types generated by our models in D.1 and an ablation on cross-lingual QA performance, finding that, compared to its English QA performance, QWEN2.5-72B-IT exhibits substantial room for improvement in cross-lingual QA (see Appendix D.4).

LLMs to directly generate comprehension questions targeting deeper cross-lingual semantic fidelity, directly conditioned on candidate and ground-truth texts.

**LLM-as-a-judge** Recent approaches utilizing LLMs as evaluative judges have emerged across NLP tasks (Liu et al., 2023; Shen et al., 2023; Kim et al., 2024a; Hada et al., 2024; Vu et al., 2024), typically producing implicit quality assessments. For MT, Kocmi & Federmann (2023b) use LLMs for holistic judgments, whereas Fernandes et al. (2023) employ detailed error assessments. Concurrently, Shafayat et al. (2024) propose rubric-based questions generated by LLMs for holistic evaluation. Distinctively, our QA-driven LLM approach explicitly assesses cross-lingual comprehension, pinpointing individual inaccuracies and providing complementary, precise insights sensitive to semantic accuracy.

## 7 Practical Recommendations

Our experiments offer several practical insights for those looking to use TREQA to evaluate paragraph-level MT, particularly in complex domains like literary texts as discussed below:

- **Start with Candidate-Aware QAG:** Conditioning the generation of questions on candidate translations to be evaluated along with the source / reference texts (the "with candidates" setting) generally yields the best results, as it helps the LLM focus on discriminative differences and potential errors (Sections 4, 5, D.1).
- **Use Powerful LLMs:** Leveraging the most capable LLMs available for both Question-Answer Generation and Question Answering is crucial. Although there is no single best model across all scenarios, models like GPT-4O, SONNET-3.5, and QWEN2.5-72B-IT consistently perform well (Table 1, Figure 3).
- **Always Regenerate Answers:** Re-generating answers using the QA model, grounded in the source/reference text, is essential to mitigate strong positional biases and hallucinations and ensure that the evaluation truly reflects comprehension of the ground truth rather than copying from candidates (Figure 5).
- **Consider Sampling for QAG:** While greedy decoding offers consistency, using sampling (e.g., temperature > 0) during question generation can explore a wider range of potential issues and often leads to higher mean accuracy over multiple runs, especially with larger models (Figure 3).
- **Non-English Target Texts:** The current TREQA framework relies on English questions and shows reduced QA performance when the context is not in English (D.4). Hence for evaluating translations *into* languages other than English, we would recommend leveraging newer multilingual LLMs with stronger cross-lingual QA capabilities with TREQA.
- **Iterate and Inspect:** The questions generated by TREQA offer interpretability. Manually inspecting the questions and answers can provide insights into why a translation is favored and can help refine prompts or settings for specific use cases.

## 8 Conclusion

Our work underscores the potential of pragmatic and extrinsic evaluation approaches such as TREQA to assess the quality of translation of complex, long-range contexts found in literary domains. By leveraging LLM-generated comprehension questions that explicitly target content (key concepts, phrases, and entities) in the source or reference texts and target errors in translations and key concepts required , TREQA not only matches but often surpasses existing state-of-the-art metrics, while also providing interpretability through explicit identification of translation errors. Although challenges remain due to the limited cross-lingual capabilities of current models, our framework paves the way for future research towards more informative and practically useful translation evaluation methodologies.

## Acknowledgments

We thank Jessy Li, António Farinhas, Eleftheria Briakou, Duarte Alves, and the SARDINE lab team for helpful discussions and feedback on earlier versions of the paper. This work was supported by EU's Horizon Europe Research and Innovation Actions (UTTER, contract 101070631), by the project DECOLLAGE (ERC-2022-CoG 101088763), by the Portuguese Recovery and Resilience

Plan through project C645008882-00000055 (Center for Responsible AI), and by FCT/MECI through national funds and when applicable co-funded EU funds under UID/50008: Instituto de Telecomunicações.

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

# A  Additional Prompts

> **Prompt for Question Generation (with optional candidates)**
>
> **System:** You are an expert teacher tasked with designing reading comprehension questions.
>
> **User:** Generate question-answer pairs to verify translation accuracy. Each answer should be a key phrase, concept, or entity from the original passage that is important in context and could help detect potential errors or mistranslations in the alternatives.
> The questions and answers must be strictly in English while ensuring that the meaning of the answer is preserved. The questions should be diverse and cover different aspects of the passage.
> Answer in the format:
> Q: <question1>
> A: <answer1>
> Q: <question2>
> A: <answer2>
>
> **Original Passage:** {src_passage}
>
> **Alternative Passages:** {alternatives}
>
> **Question-Answer Pairs:**

Figure 7: Prompt used for generating question-answer pairs in the *reference-free* setting. Text in **gray** is optional.

# B  Metric Baselines

We compare against the following metrics:

- CHRF (Popović, 2015): A character n-gram-based F-score metric that measures translation similarity. It has been demonstrated to be a strong baseline, outperforming lexical metrics such as BLEU (Mathur et al., 2020).
- COMET (Rei et al., 2022a): A reference-based neural metric based on XLM-Roberta (Conneau et al., 2020), trained on a large collection of *direct assessments* of translation quality collected over the past decade as a part of the WMT evaluation campaigns.
- COMET-QE (Rei et al., 2022b): A reference-free version of COMET based on InfoXLM-R (Chi et al., 2021) that predicts a quality score given a source text and its translation. Similar to reference-based COMET, it is trained on direct assessments.
- XCOMET (Guerreiro et al., 2024): An improved version of COMET, leveraging larger XLM-Roberta backbones and trained to predict both sentence-level direct assessments of translation quality and token-level MQM error span annotations. We compare against both the reference-based and reference-free variants.
- GEMBAMQM (Kocmi & Federmann, 2023a): An LLM-based metric that uses a fixed three-shot prompt to detect MQM error spans and compute a quality score in a reference-free manner. We use the same LLM backbone as TREQA for a fair comparison.
- MTEQA (Krubiński et al., 2021): A *system-level* QA-based evaluation metric that uses keyphrases extracted from the reference to generate questions and computes a lexical score comparing the answers to those generated using MT candidates.

# C  Additional Details for Ablation Analysis

In this part, we provide additional details on the analysis conducted in Section 5, regarding coverage of translation errors. As mentioned in the main text, we use two complementary metrics: (I) exact coverage via recall and (II) partial coverage using CHRF (Popović, 2015). Particularly, we compute:

- (I) the exact coverage via recall, as the total number of error spans covered by at least one of the available answers, over the total number of error spans (in the entire dataset).

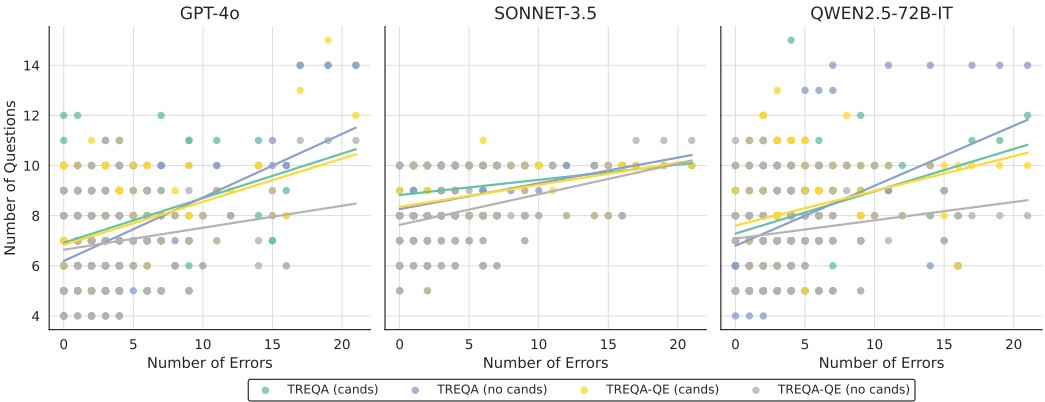

Figure 8: Number of questions vs. number of errors for all models: TREQA (no cands) achieves the highest correlation with number of errors.

- (II) the partial coverage (PC) using CHRF, by measuring the maximum similarity of each error span with the available answers of the $k$-th passage, $sim_i^k = \max_{a \in A^k} \text{chrF}(a, e_i^k)$ and then averaging over the total number of annotated error spans in the entire dataset, $PC = \frac{\sum_{k=1}^{K} \sum_{i=1}^{N^k} sim_i^k}{\sum_{k=1}^{K} N^k}$, where $A^k$ denotes the set of candidate-answers of the $k$-th passage, $e_i^k$ the $i$-th error span of the $k$-th passage, $N^k$ the number of error spans in the $k$-th passage, and $K$ the total number of passages.

## D Additional Ablations

### D.1 Type and Number of Questions

We report the number of questions against the number of expert-annotated errors across all settings and models in Figure 8. Interestingly, the candidate-agnostic, reference-based variant of TREQA exhibits the strongest correlation. Close behind are the candidate-aware QAG models, both the reference-based (TREQA) and the reference-free (TREQA-QE) variants. In contrast, the candidate-agnostic QE variant consistently lags behind—except in the case of SONNET-3.5, where it performs comparatively better. These findings suggest that access to the reference provides a strong and useful signal for identifying potential errors even for relatively weak models, while relying on the source text alone may not offer sufficient cues for effective error detection.

In Figure 9, we present an analysis of the types of questions generated by all TREQA models, categorized according to the ontology proposed by prior work (Cao & Wang, 2021). To ensure consistency and reliability in question type generation, we prompt GPT-4O using the instruction template provided and validated by Trienes et al. (2024). Across all TREQA variants, we observe a consistent distribution in question types: questions assessing conceptual understanding and the consequences of actions or events dominate across models. This trend suggests that the models effectively follow the given instruction, successfully generating questions that probe key ideas and implications presented in the passages.

A more fine-grained comparison reveals that candidate-aware variants of TREQA generate a relatively higher proportion of concept- and consequence-based questions, while producing fewer judgmental questions, those that require subjective or personal interpretation from the answerer. This shift indicates that incorporating candidate information steers the models toward generating questions that are more grounded in the passage content, reducing reliance on open-ended or opinion-based prompts. Such behavior further supports the utility of candidate-aware generation in focusing question types toward more diagnostic and error-relevant content.

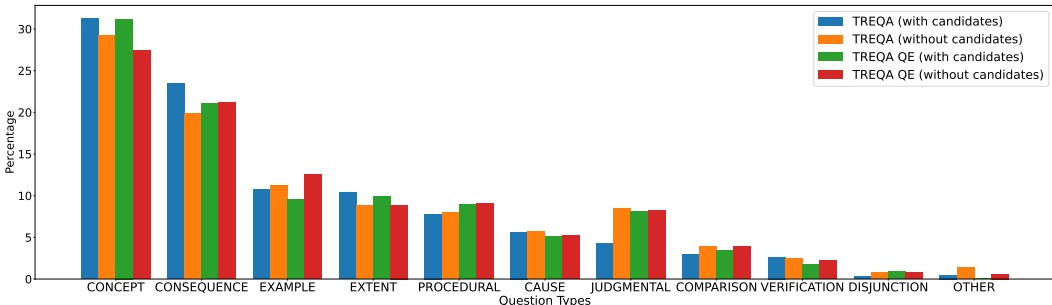

Figure 9: Distribution of question types for all models using ontoglogy defined by Cao & Wang (2021).

| QA System | Cands? | LITERARY | PAR3HUMAN | | | |
| | | TREQA | TREQA-QE | DE→EN | FR→EN | RU→EN |
|---|---|---|---|---|---|---|
| QWEN2.5-72B-IT | MULTI | **64.4** | 46.7 | 45.3 | 46.7 | 57.3 |
| | SINGLE | 60.0 | 54.4 | 43.3 | 43.3 | 57.3 |
| QWEN2.5-7B-IT | MULTI | 61.1 | **55.0** | **49.3** | 42.7 | 52.7 |
| | SINGLE | 62.8 | 53.3 | 43.3 | 43.3 | **60.0** |

Table 2: Impact of using candidate-aware generation when using single vs multi-candidate question generation and different QA systems on Contrastive Accuracy across datasets.

## D.2 Is TREQA useful in a single-candidate setup?

While TREQA by default conditions the generation of questions simultaneously on all candidates available in the first step, we also explore an alternative approach in which question generation is conditioned upon **a single candidate at a time**. This alternative enables a closer examination of the specific errors of each candidate, and pooling the questions generated by multiple candidates can facilitate comparative analysis. However, performing such a comparative analysis requires making $N$ separate calls to the question generation module – one for each candidate being evaluated – which makes this method less efficient. Table 2 reports the contrastive accuracy obtained on all evaluated datasets (LITERARY and PAR3HUMAN) when using SINGLE vs MULTI candidate-aware question generation. While MULTI on average outperforms SINGLE as it leverages comparative context across candidates to produce more discriminative questions, the results show that the questions generated conditioned on individual candidates can also provide a weaker but useful signal when ranking alternative passages.

## D.3 How does a weaker QA system paired with a strong QAG performs?

Table 2 shows the contrastive accuracy on the evaluated dataset using a relatively smaller QA system (QWEN2.5-7B-IT). As hypothesized earlier, QWEN2.5-7B-IT performs competitively with QWEN2.5-72B-IT, and in most QE settings even better. We speculate that smaller models like QWEN2.5-7B-IT may be better suited to handling straightforward factual questions, especially those that rely on surface-level cues or lexical mismatches. In contrast, QWEN2.5-72B-IT shows clear gains when paired with candidate-aware question generation, where questions often demand deeper reasoning or nuanced error interpretation.

## D.4 How well do current LLM-based QA systems perform?

We benchmark the performance of the QWEN2.5-72B-IT QA system on human-authored English questions sourced from the XQUAD dataset (Artetxe et al., 2020), using prompts described in Section 2. To assess answer correctness comprehensively, we evaluate responses using contexts in both English (denoted as QUIP) and non-English languages (denoted as QUIP (XX)). Additionally, we report several context-agnostic metrics commonly employed in QA evaluations, including CHRF, F1,

| PROMPT | CONTEXT | QUIP (EN) | QUIP (XX) | CHRF | F1 | LR |
|---|---|---|---|---|---|---|
| Extractive | EN | 4.76 | - | 81.78 | 0.77 | 3.083 |
| Crosslingual | EN | 4.41 | - | 56.20 | 0.46 | 6.851 |
| | DE | 4.16 | 4.18 | 46.19 | 0.36 | 8.172 |
| | RU | 3.99 | 4.10 | 42.95 | 0.32 | 8.686 |
| | ZH | 4.22 | 4.22 | 46.49 | 0.36 | 8.085 |

Table 3: Several answer correctness metrics for QA Evaluation using QWEN2.5-72B-IT.

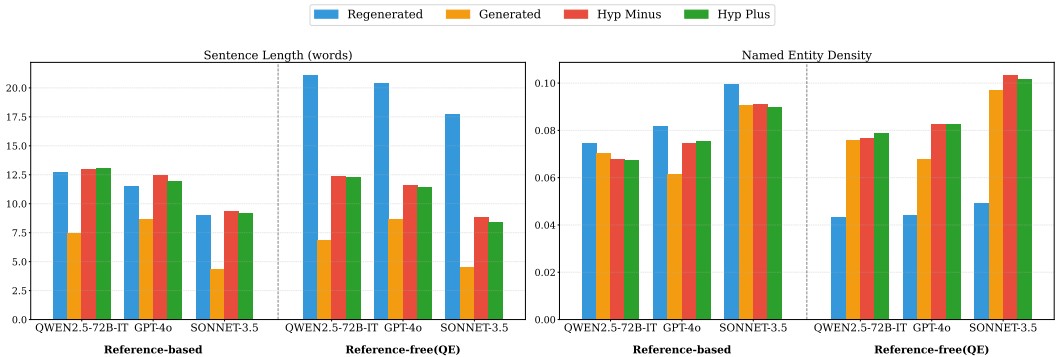

Figure 10: Answer complexity measures for TREQA when using cands.

and Length Ratio (LR), detailed in Table 3. Our findings demonstrate that the EXTRACTIVE prompt notably outperforms the CROSSLINGUAL prompt in English contexts. Specifically, using the latter results in significant performance degradation, evidenced by a sharp decrease in CHRF and F1 scores by 25.58 and 0.31, respectively. This drop in performance is accompanied by an increased Length Ratio (LR), reflecting higher complexity and reduced precision in generated responses. Furthermore, performance in cross-lingual QA tasks consistently falls behind the English-language baseline across all evaluated languages. These results highlight the difficulties in effectively achieving cross-lingual QA capabilities. While improved prompt engineering could partially address this issue, we believe that improved cross-lingual/multilingual pretraining remains crucial to further progress.

### D.5 How complex are the answers obtained from the QAG/QA system?

We assess the linguistic complexity of answers by examining both the regenerated and original reference answers, as well as the answers obtained from the candidate passages. Specifically we quantify answer complexity by measuring the total number of words and the density of name entities[10] present in each sentence. Figure 10 presents the results. Crucially, the regenerated reference answers produced by the QA system, exhibit significant differences between the reference-based and reference-free (QE) metrics in terms of both length and named entity density. Particularly, we observe that across reference-free metrics, the regenerated answers are characterized by excessive verbosity and a minimal amount of named entities, when compared to the original reference answers – those produced from the QAG system – and those derived from the candidate passages. On the contrary, under reference-based metrics, the regenerated answers are considerably less verbose and more similar to candidate-derived answers, regarding both length and the inclusion of named entities. These findings, in conjunction with the observations discussed in the preceding paragraphs, underscore the challenges associated with achieving effective cross-lingual capabilities—especially in scenarios where access to the reference passage is unavailable.

---

[10]Measured by the spaCy toolkit: https://github.com/explosion/spaCy.

