# OpenReview forum: "Do LLMs Understand Your Translations? Evaluating Paragraph-level MT with Question Answering"
_colmweb.org/COLM/2025/Conference — COLM 2025_

### Official Review · Reviewer_oatu · 2025-05-09

**Rating:** 5
**Confidence:** 4
**Ethics Flag:** 1

**Summary:**

This work proposes a new metric to evaluate paragraph-level machine translation. Specifically, TREQA, is an extrinsic metric that evaluates the quality of translations using question-answering. The metric relies on using LLMs to generate questions, and also to check for answer consistency between the source and the translation. The proposed metric is meta-evaluated on two dataset for paragraph-level MT.

**Questions To Authors:**

1. Line 216 – something more actionable on this would be good. If one is using this metric today, how should one select the model for question-generation? Do we always need some expert preferences to do model selection?

2. Footnote 8: ‘depending the’ → ‘depending on the’

**Reasons To Accept:**

1. One of the main advantages of the proposed metric is that it is completely unsupervised — it does not require any human judgements to train the metric and only relies on consistency of the answer.

2. The paper considered multiple source languages (Polish, French, Japanese, Russian etc.) which is great.

3. The qualitative error analysis to understand where and how the metric fails (line 273 and Figure 6) is really useful.

4. The set of baselines used in the main experiment and the range of LLM-backbones considered seem substantially large and diverse.

**Reasons To Reject:**

1. Conceptually unclear if using QA to evaluate translation makes as much sense as using QA to evaluate other things such as factuality (which prior work did). For translation, it is possible that the candidate contains errors (e.g. grammatical errors, only important entities are translated correctly etc) in a way that QA can still work while translation quality being poor. In general whether this framework makes sense would critically depend on the kind of questions generated — from what I understand, most questions are ‘local’ but what is really required for paragraph-level MT is questions requiring information across multiple sentences. Currently there is no analysis / evidence to support that questions have this property.

    a. For example, if a translation of an English sentence contains ‘Paris’, one might still answer ‘where’ questions correctly from a poor translation. But more complex questions would be harder to answer.

    b. For things like evaluating factuality of summaries (which prior work did) such ‘local’ questions make a lot of sense.

    c. More broadly, I’m concerned that the metric the way it is implemented right now might be easy to ‘game’ — get a high score for poor translations.

    d. Maybe combining intrinsic and extrinsic evals could be a good idea?

2. Overall the meta-evaluation results in Table 1 also doesn’t always seem great. For Literary dataset since random chance is 50%, the performance doesn’t seem that much better especially since there are only 180 examples (line 174). Some statistical significance tests could be useful. The proposed metric also doesn’t perform that great on the Par3Human dataset.

3. Some of the claims also seem to be generalized incorrectly – e.g. line 224 states including candidates reduces variance but that is only true for 1 out of 2 dataset — I don’t know if that’s a strong enough signal to make this conclusion.

4. Only the xx-en translation direction is considered now. While the paper does mention a reason for this (can use existing paragraph-level translation judgements for this direction), I think the fact that LLMs are much better at English than other languages makes it unclear if this method will generalize to the other direction.

5. (minor) The general idea of using QA to evaluate translations (or using QA to evaluate other things like factuality in summarization) has been around for a long time, so the main novelty could be somewhat limited — using LLMs to generate the questions seems to be the main difference.

---

> ### Author Response · Authors · 2025-06-02
> **Response**
>
> > “Conceptually unclear if using QA to evaluate translation makes as much sense as using QA to evaluate other things such as factuality (which prior work did)”
>
> We appreciate the reviewer’s feedback and the opportunity to clarify. While QA-based evaluation has been used in other tasks such as summarization, we argue that factuality—especially at the document level—is also crucial for machine translation (MT), and remains underexplored in the MT evaluation literature: existing metrics such as BLEU and COMET are often limited in their ability to capture these phenomena.
>
> However, to clarify, our questions **are not** restricted to simple factual or entity-centric forms. As shown in Figure 9, only about 30% are concept-based; the rest span procedural, judgmental, and consequence-based phenomena. Many questions, such as the one in Figure 6 (bottom-left), require reasoning across sentences, demonstrating that the QA signal is not purely local:
>
>  “How is the cousin related to Grandma’s litte sister”?
> requires understanding
> ```My father, in fact, had no idea how to do business, but he had gone to seek refuge with a distant cousin. **My grandmother had a younger sister** who had gone to Beijing in her youth, and had gotten married and had children there. **This cousin was her eldest son**, and had always been an outcast in the family. ```
>
> TREQA is designed to surface translation errors regardless of granularity—be they lexical or discourse-level—and we demonstrate in Table 1 that it performs competitively with state-of-the-art MT metrics. Moreover, Section 5 shows that TREQA questions align well with human-identified errors, further validating their diagnostic value.
>
> Finally, we agree that combining extrinsic QA-based signals with intrinsic metrics (e.g., COMET) may offer complementary strengths, particularly for nuanced paragraph-level evaluation. We will clarify this motivation and its connection to factuality evaluation in other NLP tasks in the final version.
>
> > On TREQA’s performance:
> We agree that TREQA is not always the top-performing metric across all datasets. However, it's important to contextualize these results. Even the strongest reference-based metrics, such as COMET and XCOMET—both of which are trained on large amounts of human judgments—or GEMBA-MQM, which is explicitly designed to target error spans, perform in a similar range. TREQA, despite not being supervised on human judgments and operating in a more general QA-based framework, achieves competitive performance, which we believe is a notable strength.
> Regarding the Literary dataset specifically, we acknowledge that the performance margin above random (50%) may not appear large. However, this dataset is particularly challenging due to its domain complexity and the small size (180 examples), which inherently limits statistical power. We agree that conducting statistical significance tests would help clarify the robustness of the observed differences, and we consider this an important direction for future analysis.
>
> Finally, it should be noted that the goal of our paper is not to propose a metric to replace existing ones, but rather to develop a metric which captures in a more natural and interpretable way phenomena which existing metrics do not capture well -- we believe our metric can be used in tandem with other metrics such as COMET, and therefore we do not aim to always outperform existing metrics (specially metrics which are specifically trained to correlate with human judgments)."
>
> > On Variance Reduction:
> Please note that including candidates reduces variance in 4/6 cases (across model sizes and the two LPs considered). Having said that, we agree that the claim could be better justified -- we will make it more specific to the results in the header paragraph. Thanks for pointing that out.

---

> > ### Author Response · Authors · 2025-06-02
> > **Response Continuation**
> >
> > > On Generalizability to non-english languages:
> > Yes, our current experiments focus solely on the xx→en direction. However, this was a deliberate choice, primarily driven by the availability of high-quality paragraph-level human judgments for this translation direction, which were essential for validating our metric (see footnote 5 in the paper for other reasons). However, TREQA does leverage cross-lingual context in a reference-free setup, which at least partially mitigates concerns about language bias. Specifically, both question generation and answering are informed by the source input and candidate translation, allowing the model to reason across languages rather than relying solely on English fluency.
> >
> > We acknowledge that English-centric LLMs may perform better when evaluating English outputs, and that generalizing this approach to the en→xx direction—or other non-English target languages—poses an important challenge. However, we view this work as a first step and proof of concept. The core framework of using QA for MT evaluation is language-agnostic in principle, and future work could incorporate multilingual LLMs or language-specific QA models to extend the methodology to other directions.
> >
> > > ”The general idea of using QA to evaluate translations (...) has been around for a long time, so the main novelty could be somewhat limited”
> > We agree that the idea of using QA as an evaluation mechanism—whether for summarization, translation, or factuality—has been explored in prior work. However, the key contribution of our work lies in how we operationalize this idea specifically for machine translation evaluation in a way that meaningfully extends prior approaches.
> >
> > Our main contributions include: 1) candidate-aware question generation using LLMs, which tailors the questions to the actual translations being evaluated. 2) applicability to paragraph-level MT, where traditional sentence-level metrics often fall short. 3) comprehensive empirical analysis showing that our QA-based metric, TREQA, is competitive with state-of-the-art learned metrics, despite not relying on large-scale human judgment supervision.
> >
> > So while the general idea of QA-based evaluation is not new, we believe our specific implementation — especially leveraging LLM capabilities for dynamic, candidate-aware question generation — represents a meaningful and novel contribution.
> >
> > > On selecting one configuration:
> > To ensure the best possible performance of TREQA, we recommend selecting best setting via a *validation* set of human preferences (as different LLMs might be better at generating questions/answers in different domains). However, our findings also hint that using a strong frontier LLM like Claude/GPT always yields a strong evaluator: for example using GPT4o for QG surpassed XCOMET on 4/5 settings we experimented with. However, we agree that more actionable claims would be beneficial for readers, and will make sure to add this to the paper.
> >
> > We hope that our answers alleviate our main concerns and, if that is the case, we kindly ask you to reconsider your recommendation score. We are happy to address any remaining questions you may have.

---

> > > ### Comment · Reviewer_oatu · 2025-06-05
> > >
> > > Thanks for the detailed reply!
> > >
> > > 1. " we argue that factuality—especially at the document level—is also crucial for machine translation (MT), and remains underexplored in the MT evaluation literature:" --> while I definitely agree with this, my thought was that evaluating MT holistically requires a lot more than just evaluating factuality. (In prior work like summarisation extrinsic QA evaluation was only used for factuality). Hence the suggestions for combining with intrinsic eval.
> > >
> > > 2. Thanks for pointing out the figure 6 example! It could be useful to add more analysis to show how many such questions are generated to make the paper stronger.
> > >
> > > 3. The concern regarding variance reduction and xx-en direction seems sufficiently addressed. Thanks!
> > >
> > > 4. The performance being close to random chance and missing significance testing is still a weakness. The novelty concern also remains.
> > >
> > > Overall especially considering point 4 above, I'm happy to increase score but would still prefer to keep it marginally below acceptance.

---

> > > > ### Author Response · Authors · 2025-06-10
> > > > **Thank you for the response.**
> > > >
> > > > 1. We agree that combining it with intrinsic evaluation would be interesting, however it is an open question to devise a good strategy to approach this (for e.g. we can aggregate the error spans obtained by the two with some careful considerations or aggregate (average, weighted scoring) the scores after some normalization or combine the rankings induced by the two approaches or even multiple metrics) and resolving this is beyond the scope of our work. We will add a sentence to our conclusion to explicitly highlight this as a promising direction for future work.
> > > >
> > > > 2.We will add more examples to the paper. We note that a summary of type of questions is currently already present in Figure 9 of the appendix.
> > > >
> > > > 3. We are glad we were able to resolve this.
> > > >
> > > > 4. We acknowledge the reviewer's concern about the lack of significance testing. To address this, we will incorporate permutation tests in the final version of the paper [1]. We understand the reviewer's point about the general concept of QA for evaluation. However, we respectfully reiterate that the core novelty of TREQA lies in its specific implementation tailored for MT. The candidate-aware question generation is our core novel aspect and grounding question generation on all candidates being evaluated/compared has not been tested in prior work (to the best of our knowledge). And, TREQA is explicitly designed and demonstrated to be effective for paragraph-level MT evaluation - which is the direction we are moving as a community [2]. Finally, it is also important to highlight that TREQA doesn't require any training and still remains competitive with state-of-the-art neural metrics.
> > > >
> > > > [1] Thompson, B., Mathur, N., Deutsch, D., & Khayrallah, H. (2024). Improving Statistical Significance in Human Evaluation of Automatic Metrics via Soft Pairwise Accuracy. arXiv preprint arXiv:2409.09598.
> > > >
> > > > [2] Post, Matt, and Marcin Junczys-Dowmunt. "Escaping the sentence-level paradigm in machine translation." arXiv preprint arXiv:2304.12959 (2023).

---

### Official Review · Reviewer_unEd · 2025-05-10

**Rating:** 7
**Confidence:** 4
**Ethics Flag:** 1

**Summary:**

The paper presents a framework to evaluate MT using QA, with questions generated from elements in the source or reference translations. It addresses the problem that MT evaluation metrics are still mostly at the sentence-level.

The framework uses LLMs to generate questions that target the content of either the source or reference texts. These questions are conditioned on the set of translation hypotheses, so they focus on the differences between the hypotheses. The framework then evaluates how well answers generated from each translation align with those derived from the source or reference texts.

The evaluation always concerns the into English direction, but context can be non-English.

The results are mixed but generally positive and competitive.

I really like that the paper is a serious attempt at an extrinsic evaluation metric for MT. However, there are definitely discussion points related to the method (see questions below). In addition, clarity can be improved. I had some trouble understanding some central points like translation directions for the QA and where and how the question-answer pairs are being generated.

**Questions To Authors:**

- are hallucinations a concern in the QA generation step?

- in section 2.2, lines 155-162, I found the explanation of the answer regeneration somewhat unclear. Is it the case that both an LLM and a separate QA system are used to generate answers? If yes, does this mean that the questions and answers are produced by different systems or is it that different models integrated within the QA process? This becomes somewhat clearer in section 5 (I think), but I'd recommend to clarify this part earlier in the text. Basically, I'd offer a very simplified explanation of the method parts in the introduction. On the same topic, about paragraph at line 262, could you elaborate on what do you mean "true comprehension" in this context?

- what's the reasoning for not including a document-level metric among the baselines?

- looking at table 1, it seems that GembaMQM is somewhat better, what is TREQA's added value?

- In section 5, paragraph at line 246, are these answers that include an error and explicitly say there's an error in the translation? As a potential user, how do you know the translation has an error?

Small stuff:

- Figure 6 is difficult to read when the paper is printed

**Reasons To Accept:**

- proposal to do extrinsic evaluation of MT that is ambitious and new

- strong potential to evolve into a more robust and comprehensive framework with further development

**Reasons To Reject:**

- Potential concerns regarding methodological choices (some could be a matter of clarification):

	i) the idea is to evaluate if people with access to the translations have the same understanding as people with access to the sources. So why choosing literature as test? Wouldn't a genre with more objectively measurable success/fail outcomes, for example instructions, be better suited? The literary genre seems open to interpretation and hence as confounding factor.

	ii) document-level metrics for MT seem relevant but missing

---

> ### Author Response · Authors · 2025-06-02
> **Response**
>
> > “why choosing literature as test? Wouldn't a genre with more objectively measurable success/fail outcomes (...) be better suited?”:
>
> Thank you for raising this important point. In an ideal setting, a truly extrinsic evaluation would require taking actions based on the translations and evaluating quality of the translations based on the success of those actions [1, 2]. However practically this is very challenging and is difficult to scale. Our work takes a more pragmatic yet informative approach by using QA as a proxy for downstream comprehension [3]. This allows us to assess whether readers of the translation extract the same key information and inferences as readers of the original, without requiring complex task-based setups.
>
> We also believe that the literary domain remains a relevant and challenging benchmark for several reasons as these texts demand deep semantic understanding, multi-sentence reasoning and subtle discourse enforced meaning nuances that traditional metrics would fail to capture. While literature is open to interpretation, many questions generated by TREQA focus on elements such as character relationships, event sequences, and causal links, which have stable, answerable properties across translations. Furthermore, by grounding the questions on the candidates, we ensure that the question are relevant in surfacing translation errors which is the main goal of TREQA.
>
> We will add this discussion in the introduction to strengthen the motivation for our choices for this work.
>
> [1] Church, Kenneth W., and Eduard H. Hovy. "Good applications for crummy machine translation." Machine Translation 8 (1993): 239-258.
> [2] Doyon, Jennifer, Kathryn B. Taylor, and John S. White. "Task-based evaluation for machine translation." Proceedings of Machine Translation Summit VII. 1999.
> [3] Tomita, Masaru, et al. "Evaluation of MT Systems by TOEFL." Proceedings of the Fifth Conference on Theoretical and Methodological Issues in Machine Translation of Natural Languages. 1993.
>
> > ”document-level metrics for MT seem relevant but missing”:
>
> Please note that all the metrics reported in Table 1 are operationalized at the paragraph level, which aligns with common practice in recent machine translation evaluation research.. While some metrics have been proposed as document-level, they often operate in a context-aware fashion—typically by processing texts at the sentence or segment level and incorporating limited surrounding context (e.g., via context windows) as discussed in Section 6. Such methods often rely on tools like BleuAlign to split paragraphs into aligned segments or make assumptions about the linear one-to-one sentence-level ordering in translation, potentially losing crucial cross-sentence information.  In our work, we intentionally avoid such segmentation in order to preserve the full paragraph structure and better reflect real-world paragraph-level translation quality.
>
> However, if there are other document-level metrics that we may have overlooked, we kindly invite the reviewer to share relevant citations. We would be happy to include a discussion or comparison in the final version of the paper.

---

> > ### Author Response · Authors · 2025-06-02
> > **Response to other questions**
> >
> > > "are hallucinations a concern in the QA generation step?"
> > yes, hallucinations can occur during question generation. To minimize this, we ground the questions directly on the candidate translations, which helps ensure that questions remain relevant and tied to the actual content. Nonetheless, some hallucinations still happen, as illustrated in Figure 6 (bottom left). We acknowledge this as an area for future improvement as also already discussed in Section 5 (lines 281-283).
> >
> > > "what do we mean by  "true comprehension"?"
> > in this context, “true comprehension” refers to model’s ability to understand and reason about the reference material (i.e., the passage or source text) rather than simply copying surface forms from the input (which could include just the reference or additionally candidates): e.g. if answer regeneration is not used, a candidate translation can receive a high score simply because the “gold” answer was copied directly from that candidate during the QAG process, regardless of whether the translation meaningfully supports question answering.  However we agree that “true comprehension” is a confusing term, and will rephrase the claim to “...and thus the answers extracted are better suited to evaluate if candidate translations preserve meaning.”
> >
> > > "how does answer regeneration work?"
> > put simply, rather than using the QAG system’s generated answer, during evaluation of a candidate, the QA system extracts and answer from the candidate translation and also from the reference text. so its technically correct to say that both an QAG and QA system generate reference answers (both being LLMs), but only the latter is used when the “answer regeneration” is turned on.
> >
> > > "missing document-level metrics?"
> > we do report all metrics (CHRF, COMET) evaluated at the paragraph-level. If we missed anything specific, we are happy to add in the revision.
> >
> > > "what is the added value of TREQA?"
> > Gemba-MQM and TREQA offer complementary evaluation strategies: where Gemba-MQM offers fine-grained error insights driven by pre-defined expert-designed error typology, TREQA complements it by probing whether a model preserves the functional utility of a translation, i.e. by allowing a QA system to recover key information expressed in the reference. The latter also allows TREQA to capture errors beyond localized meaning errors (as labelled by MQM) and at the same time offer interpretable question-driven diagnostics to meaning divergences.
> >
> > >" in section 5, paragraph at line 246, are these answers that include an error and explicitly say there's an error in the translation? As a potential user, how do you know the translation has an error?"
> > We use expert-labeled ground truth MQM error span annotation to show that answers target errors. A user can know if there is an error in the translation if the answer derived from the source diverges from the candidate translation.
> >
> > > "improve figure 6"
> > thanks for pointing it out, we will improve the font and readability for this image.

---

> > > ### Comment · Reviewer_unEd · 2025-06-05
> > > **Satisfied with responses**
> > >
> > > Thank you for the detailed responses. As I suspected, some of the issues were a matter of clarification which it seems will be improved in the final version. I believe the score I initially assigned reflects the strengths of the paper, so I would prefer to keep it as is.

---

### Official Review · Reviewer_qzD2 · 2025-05-12

**Rating:** 7
**Confidence:** 4
**Ethics Flag:** 1

**Summary:**

This work proposes an alternative framework for evaluating machine translation (MT) systems through a question answering (QA) task rather than some form of direct string comparisons between the MT outputs and reference translations. The framework involves asking one set of LLMs to generate questions based on the source text (potentially with the reference translation and candidate translations as additional inputs), then asking another set of LLMs to answer those questions based on each candidate translation, and finally evaluating the accuracy of the candidate answers against ground-truth answers.

An empirical study was carried out to assess the proposed framework. The evaluation is thorough: multiple alternative MTE metrics were considered, including another LLM-based metric; multiple datasets were used, covering several source languages; several experimental conditions were tried (with and without reference, with and without candidates). The results show the proposed metric TREQA to be comparable to another LLM-based metric, GEMBAMQM for the evaluative task (accuracy of matching expert preferences between translation pairs). While the evaluative task makes it possible to directly compare with a wide range of existing metrics, it feels somewhat distant from the original goal of coming up with a more extrinsic oriented evaluation task.

Additional analytical experiments were conducted to better understand the performance of TREQA, but the findings were either not too surprising (candidate-aware generation seems to help, sampling from larger models seem to help) or still vague (ways in which TREQA fail).

Aspects of the paper are not as well described. For example, the paper stated that answer correctness is measured using LERC, but we do not know how varying this choice might impact the overall success of the framework. The application of the main experimental metric (contrastive accuracy) is also not really described (a reader of the paper may not be able to recreate the procedure of the experiment).

**Reasons To Accept:**

The proposed framework is well-motivated. It would be nice to have a more robust and in-depth way to evaluate MT beyond sentence-level comparisons with reference translations.

The overall experimental methodology is thorough.

**Reasons To Reject:**

The validation of the proposed framework ultimately still seems tied to some sort of human preference at the sentence-level.

The performance of the framework seems to depend on several factors (choice of QAG model, choice of QA system, choice of an answer correctness metric), and it is not sufficiently clear how someone might make these choices for some other use scenario (where the target language is not English, for example).

---

> ### Author Response · Authors · 2025-06-02
> **Response**
>
> > “The validation of the proposed framework ultimately still seems tied to some sort of human preference at the sentence-level…”
>
> For practical purposes, we validate our framework by showing that 1) TREQA’s rankings of candidate translations align with human judgments on *paragraph-level* texts (not sentence-level), achieving accuracy scores comparable or better than state-of-the art metrics and 2) the QA pairs created by TREQA target translation-specific errors, as evidenced by their alignment with expert-labeled MQM annotations. However, we agree that a more human-centric evaluation, such as asking annotators to assess the quality of the generated questions or answer correctness, would strengthen our result, but leave that to future work.
>
> > “The performance of the framework seems to depend on several factors…"
>
> Yes, the final performance of the metric depends on several design choices, including the input to the QAG model, the QAG model itself, the QA system, and the answer correctness metric. However, the primary contribution of our work is not a fixed configuration optimized for a specific language pair, but rather the introduction of a flexible framework. TREQA is designed to be modular, allowing practitioners to adapt each component to suit their specific needs. We note that we will release the code to reproduce the results reported in the paper. Furthermore, we will also add actionable insights (best practical configuration) and recommendation to the discussion, so that practitioners have a starting point to tailor TREQA to their specific use case, including scenarios involving target languages other than English.

---

> > ### Comment · Reviewer_qzD2 · 2025-06-05
> >
> > Thanks for the response. It would be good to include a discussion on actionable insights. I look forward to reading the revised version .

---

### Official Review · Reviewer_zwtQ · 2025-05-12

**Rating:** 6
**Confidence:** 4
**Ethics Flag:** 1

**Summary:**

The paper presents a method for MT quality evaluation and quality estimation based on automatic question answering. Questions (and answers) are automatically generated from source (and reference) and the method compares whether the same questions can be answered also based on the test translations. The results should competitive performance on two datasets (Literary dataset and PAR3HUMAN).

**Questions To Authors:**

I miss technical details of the experiments, such as the number of generated questions per sentence/paragraph and distribution of QA results. And some detailed analysis of the generated questions (though I appreciate the ablation studies in Appendix D) -- e.g how detailed are the questions.

**Reasons To Accept:**

The idea itself is not novel but the novel part is using LLMs to generate the answers. The main experiments are accompanied by several additional experiments, ablation studies, and qualitative analysis.

**Reasons To Reject:**

Limited evaluation. The experiments are done on two specific datasets only. There area other resources that could have been used to provide a better and more complete picture about the performance of the method (WMT resources for evaluation and QE), including those based on direct assessment.  The evaluation is also limited in terms of languages, the target language is always EN.
No attention is paid to systems level evaluation -- but this is given the choice of datasets.

The questions are limited to those that are extractive. There are good reasons for that but doing a contrastive experiments not limited to extractive questions would be very interesting.

---

> ### Author Response · Authors · 2025-06-02
> **Response**
>
> We thank the reviewer for the thoughtful review.
>
> > On limited evaluation:
> We used the reported datasets because, to the best of our knowledge, they are the only available sources containing genuine paragraph-level texts and expert human annotations. Although WMT datasets are labeled as supporting paragraph-level evaluation, only 30% of the examples are “truly” paragraphs. Moreover, WMT evaluations are conducted independently for each translation, whereas both Par3Human and Literary datasets support contrastive evaluation, which better highlights subtle differences in translation quality—an increasingly important concern as MT systems improve [1]. However, we acknowledge the relevance of WMT and consider extending our framework to future WMT datasets, especially those that feature genuine document-level content (e.g., WMT2025). We also clarify that system-level evaluation was beyond the scope of this work as current metrics and previous QA-based approaches already showed high correlations [2]. Our focus was on segment-level quality assessment, as it allows more granular insights into MT performance.
>
> [1] Song, Yixiao, et al. "Enhancing human evaluation in machine translation with comparative judgment." arXiv preprint arXiv:2502.17797 (2025).
> [2] Krubińsk, Mateusz, et al. “MTEQA at WMT21 Metrics Shared Task”, WMT21
>
> > On language direction:
> Yes, our current experiments focus solely on the xx→en direction (L134-138) as, among other reasons, LLMs have shown superior performance in English compared to other languages (see footnote 5 in the paper). However, TREQA does leverage cross-lingual context in a reference-free setup, which at least partially mitigates concerns about language bias. Specifically, both question generation and answering are informed by the source input and candidate translation, allowing the model to reason across languages. We view our work as a first step and proof of concept. The core framework of using QA for MT evaluation is language-agnostic in principle, and future work could incorporate multilingual LLMs or language-specific QA models to extend the methodology to other directions.
>
> > On questions being extractive:
> As stated in L148–149, our decision to use extractive QA was driven by the desire to ground the QA process strictly in the given texts. This ensures a tighter alignment between question/answer pairs and the input content. While extractive, the generated questions still span diverse linguistic phenomena, as shown in Figure 9 in the appendix. We agree that moving beyond extractive QA is an interesting direction for future work; however, a key challenge is the lack of established evaluation metrics for measuring answer correctness, which is beyond the scope of this study. Our preliminary experiments (footnote 3) using an LLM as a judge for answer correctness showed a tendency to overpredict high scores, highlighting the difficulty of reliable evaluation in this setting.

---

> > ### Comment · Reviewer_zwtQ · 2025-06-05
> >
> > Thank you very much for providing the answers. I'm sticking with my original scores.

---

### Decision · Program_Chairs · 2025-07-08

**Decision:**

Accept

**Comment:**

This paper introduces TREQA (Translation Evaluation via Question-Answering), a framework for evaluating paragraph-level MT quality via question answering. Instead of relying solely on intrinsic metrics trained to mimic human judgments, TREQA employs LLMs to generate and answer comprehension questions about the source or reference texts, using the alignment between candidate and reference answers as a proxy for translation quality.

Reviewers note that revisiting extrinsic MT eval with LLM-powered QA is new and well-motivated and has strong potential. The empirical parts of the paper have been highlighted as thorough. One reviewer calls the paper a serious attempt at an extrinsic evaluation metric for MT. The paper acknowledges variance in LLM generations, issues with hallucinations, and positional bias, but provides thoughtful ablations and discussions.

Some reviewers criticize that the target language is only English. The improvement margins, especially on the small LITERARY dataset, are not always compelling, and statistical testing was absent (though promised for the final version). Some point out that the choice of literature translation is not ideal, and that the evaluation quality depends on the quality of the generated questions.

Overall, the paper introduces an interpretable approach to paragraph-level MT evaluation. Its core ideas are mostly novel, empirically grounded, and interesting to the community.